# Hydrogen Peroxide Electrochemical Sensor Based on Ag/Cu Bimetallic Nanoparticles Modified on Polypyrrole

**DOI:** 10.3390/s23208536

**Published:** 2023-10-18

**Authors:** Yanxun Guan, Fen Xu, Lixian Sun, Yumei Luo, Riguang Cheng, Yongjin Zou, Lumin Liao, Zhong Cao

**Affiliations:** 1Guangxi Key Laboratory of Information Materials & Guangxi Collaborative Innovation Center for Structure and Properties for New Energy and Materials, School of Material Science and Engineering, Guilin University of Electronic Technology, Guilin 541004, China; 15185128385@163.com (Y.G.); luoym@guet.edu.cn (Y.L.); chengriguang@guet.edu.cn (R.C.); zouy@guet.edu.cn (Y.Z.); liaolumin0827@gmail.com (L.L.); 2School of Electronic Engineering and Automation, Guilin University of Electronic Technology, Guilin 541004, China; 3Hunan Provincial Key Laboratory of Materials Protection for Electric Power and Transportation, Changsha University of Science & Technology, Changsha 410114, China; caoz@csust.edu.cn

**Keywords:** H_2_O_2_ sensors, polypyrrole, electropolymerization, electrodeposition, Ag nanoparticle, Cu nanoparticle

## Abstract

Due to the strong oxidizing properties of H_2_O_2_, excessive discharge of H_2_O_2_ will cause great harm to the environment. Moreover, H_2_O_2_ is also an energetic material used as fuel, with specific attention given to its safety. Therefore, it is of great importance to explore and prepare good sensitive materials for the detection of H_2_O_2_ with a low detection limit and high selectivity. In this work, a kind of hydrogen peroxide electrochemical sensor has been fabricated. That is, polypyrrole (PPy) has been electropolymerized on the glass carbon electrode (GCE), and then Ag and Cu nanoparticles are modified together on the surface of polypyrrole by electrodeposition. SEM analysis shows that Cu and Ag nanoparticles are uniformly deposited on the surface of PPy. Electrochemical characterization results display that the sensor has a good response to H_2_O_2_ with two linear intervals. The first linear range is 0.1–1 mM (R^2^ = 0.9978, S = 265.06 μA/ (mM × cm^2^)), and the detection limit is 0.027 μM (S/N = 3). The second linear range is 1–35 mM (R^2^ = 0.9969, 445.78 μA/ (mM × cm^2^)), corresponding to 0.063 μM of detection limit (S/N = 3). The sensor reveals good reproducibility (σ = 2.104), repeatability (σ = 2.027), anti-interference, and stability. The recoveries of the electrode are 99.84–103.00% (for 0.1–1 mM of linear range) and 98.65–104.80% (for 1–35 mM linear range). Furthermore, the costs of the hydrogen peroxide electrochemical sensor proposed in this work are reduced largely by using non-precious metals without degradation of the sensing performance of H_2_O_2_. This study provides a facile way to develop nanocomposite electrochemical sensors.

## 1. Introduction

Hydrogen peroxide (H_2_O_2_) is an industrial raw material widely used in chemical, medical, textile, food, military, and other fields. It is also the product of many oxidase-catalyzed reactions in the human body and plays an important role in the normal physiological environment of the human body [1,2]. However, due to the strong oxidizing properties of H_2_O_2_, excessive discharge of H_2_O_2_ will cause great harm to the environment. Excessive amounts of H_2_O_2_ in the body can also bring about various diseases. Therefore, the accurate detection of H_2_O_2_ plays an important role in environmental protection, food safety, medicine, health, and other fields and has broad prospects [3,4].

There are several techniques for detecting H_2_O_2_, such as titration [5], chromatography [6], spectroscopy [7,8], and electrochemical methods [9,10]. Various methods of detecting H_2_O_2_, such as electrochemical detection of H_2_O_2_, can be used to detect the concentration of H_2_O_2_ at μM or even nM levels in solution and can also be quickly and effectively determined under harsh conditions. The choice of sensitive material is one of the most important factors in the construction of an electrochemical sensor. At present, the sensitive materials of electrochemical H_2_O_2_ sensors can be divided into two kinds: enzymatic and non-enzymatic. Although enzyme-based sensors have been found to have good sensing performance, their shortcomings, such as high cost, easy inactivation of enzymes, and susceptibility to various parameters such as pH and temperature, affect their performance in practical applications [11,12]. Thus, it is feasible and necessary to develop novel non-enzymatic electrochemical sensors.

Some nanozymes integrated with polymers have been used for sensing detection, and good results have been achieved. For example, Wang et al. used the co-precipitation method to fix NiPd hollow nanoparticles and glucose oxidase (GOx) on zeolite imidazolate acid skeleton 8 (ZIF-8) at the same time, and the prepared GOx@ZIF-8 (NiPd) nanoflowers not only showed the peroxidase-like activity of NiPd hollow nanoparticles, but the glucose was also detected by optical colorimetry [13]. Baretta et al. prepared Prussian Blue nanoparticles (PB NPs) in a cellulose-based hydrogel network under mild synthetic conditions in the presence of glucose oxidase and fixed PB NPs and active GOx in the hydrogel at the same time. The prepared electrochemical sensor could detect H_2_O_2_ and glucose at very low concentrations [14]. Park et al. used bovine serum albumin as a nucleation template and stabilizer to prepare a platinum nanozyme-hydrogel composite with high specific peroxidase-like activity. The prepared sensing unit was embedded within a restricted detection region of a plastic chip with a 3D hydrophilic fluid path to produce an efficient platform for the independent detection of glucose. The recovery of serum, urine, and saliva samples was as high as 83–105%, with high specificity for glucose and no significant interference from other sugars, and had good long-term stability and repeatability after two months of storage [15]. These methods provide hope for the development of multi-enzyme systems and establish the possibility of cooperation between artificial enzymes and natural enzymes, which is expected to achieve applications.

Polypyrrole (PPy) has attracted great interest for potential applications in batteries, supercapacitors, microwave shielding, and sensors due to its high electrical conductivity, environmental nontoxicity, and reversible redox properties [16,17,18]. However, pure PPy may have drawbacks such as low sensitivity, poor selectivity, and susceptibility to interference that prevent its commercial development. Some transition metal nanoparticles have been widely used and applied in the field of sensors due to their high catalytic activity for many chemical reactions [19]. Precious metal nanoparticles (such as Au [20,21,22,23], Ag [22,23,24,25], Pt [26,27], and Pd [28,29]) and transition metals (like Co [30], Fe [31], and Cu [32]) were composited with PPy and used in H_2_O_2_ sensing research. Meanwhile, researchers showed that the sensing effect of bimetallic doping was better than that of single metal doping due to the synergistic effect between the two metal nanoparticles [33,34]. Due to the easy modification of PPy, metal particles are easily attached to the surface of PPy, so PPy combined with bimetals can enhance the sensing effect of sensitive materials.

Generally, the sensitive material was covered on the surface of the GCE by drip coating, which would increase the risk of the sensitive material falling off to a certain extent. Compared with dripping the PPy polymerized in liquid phase onto the surface of GCE, the direct electropolymerization of PPy onto the GCE has better conductivity, and the density and thickness of the film formed on the surface of the electrode can be better controlled [35]. In addition, PPy can be firmly adsorbed on the electrode, which provides the possibility for the repeated use of the modified electrode and the improvement of endurance.

In recent years, there have been some reports on the use of Ag and Cu as PPy modification materials for H_2_O_2_ detection. The excellent catalytic activity of hydrogen peroxide reduction is one of the important properties of silver nanoparticles (AgNPs). The electrochemical data have shown that the presence of silver nanoparticles was the reason for the high sensitivity of the modified electrode to H_2_O_2_ reduction [36]. The modification of copper could not only increase the electrical conductivity of the material but also enhance the sensing performance of the material [32]. Electrodeposition methods for Ag and Cu have also been reported. For example, Hoang et al. used the constant current method to co-deposit Ag and Cu on carbon paper to obtain a high-specific surface area alloy film that had good electric reduction performance for CO_2_ [37]. Bernasconi et al. used pyrophosphate-iodide electrolyte to electroplate Ag–Cu alloy, using a copper plate as an electrode and electrodeposition at 50 °C. They explored the principle of electrodeposition of Cu and Ag and the effect of using different electrolytes on the structure of the alloy [38].

In this paper, Ag and Cu are modified on the surface of PPy by electrodeposition, which conveniently and quickly solves the problem of insufficient sensitivity to H_2_O_2_ when only PPy is used as a sensitive material. Compared with the method of modifying PPy with Ag alone as a sensitive material, it not only improves the sensitivity of the sensitive material but also reduces the cost of preparation, which provides the possibility for the practical application of the sensitive material. Compared with Cu modification of PPy as a sensitive material alone, the linear range of the sensitive material is broadened, and the detection limit becomes lower, which is conducive to more extensive detection of H_2_O_2_.

## 2. Materials and Methods

### 2.1. Materials

All materials, including pyrrole (Beijing InnoChem Science & Technology Co., Ltd., Beijing, China), sodium chloride (NaCl) (Sinopharm Chemical Reagent Co., Ltd., Shanghai, China), silver nitrate (AgNO_3_) (Xilong Scientific Co., Ltd., Shantou, China), copper nitrate [Cu (NO_3_)_2_] (Sinopharm Chemical Reagent Co., Ltd., Shanghai, China), sodium phosphate monobasic dihydrate (Na_2_HPO_4_·2H_2_O) (Sinopharm Chemical Reagent Co., Ltd., Shanghai, China), sodium phosphate dibasic dodecahydrate (NaH_2_PO_4_·12H_2_O) (Sinopharm Chemical Reagent Co., Ltd., Shanghai, China) and Hydrogen peroxide (H_2_O_2_) (Xilong Scientific Co., Ltd., Shantou, China) were bought from Aladdin Reagent Inc. (Shanghai, China). The glassy carbon electrode (GCE), silver chloride reference electrode, and platinum wire electrode were purchased from Tianjin Aida Hengsheng Technology Development Co., Ltd. (Tianjin, China).

### 2.2. Characterization

Scanning electron microscopy (SEM; Quanta 200, FEI, Thermo Fisher Scientific, Waltham, MA, USA) was used to characterize the microstructures and morphologies of the samples. Microstructure analysis was also conducted by transmission electron microscopy (TEM; FEI Tecnai G2 F30, Thermo Fisher Scientific, Waltham, MA, USA). Energy dispersive spectroscopy (EDS) was performed during the scanning transmission electron microscopy (STEM) tests.

### 2.3. PPy–Ag/Cu Electrode Fabrication

Firstly, a glass carbon electrode (GCE) was used as the working electrode to prepare the H_2_O_2_ sensor. Before the experiment, the GCE was polished to mirror the surface with alumina with diameters of 100 and 50 nm, respectively, and then ultrasonic cleaned with distilled water and 50% ethanol. The treated electrode was dried under high-purity N_2_ gas.

Then, the PPy film and Ag/Cu particles were modified on GCE to obtain the PPy–Ag/Cu electrode, respectively. Briefly, PPy film was electropolymerized on the surface of GCE by the cyclic voltammetry (CV) method, and the solution used for polymerization contained 0.06 M pyrrole monomer and 0.1 M NaCl. The voltage range was set to −1.0 V–1.0 V, and the number of cycles and sweep speed were 8 and 25 mV/s, respectively. Finally, the PPy/GCE was inserted into the aqueous solution containing 2.5 mM AgNO_3_ and 5 mM Cu (NO_3_)_2_. Ag and Cu particles were also electrodeposited on its surface by the CV method. The voltage range and sweep speed were the same as above. However, the number of deposition cycles was 10. The above solutions used for electropolymerization and electrodeposition are all purged with N_2_.

### 2.4. Electrochemical Measurements

Cyclic voltammetry (CV) and timed amperometry were used to evaluate the sensing performance of PPy–Ag/Cu for H_2_O_2_ in a 0.2 M phosphate buffer solution. The CV was carried out at potentials ranging from 0 to −0.65 V with a scan rate of 50 mV/s. Timed amperometric measurements were performed at a constant voltage of −0.5 V. Before adding H_2_O_2_, the background current was stabilized to a certain constant value, and then a certain amount of H_2_O_2_ was added to the solution under stirring conditions to obtain an amperometric *i–t* curve. All experiments were performed at room temperature in 10 mL of 0.2 M phosphate buffer solution (PBS).

## 3. Results and Discussion

### 3.1. Characterization

A scanning electron microscope with an energy-dispersive spectrometer is utilized to identify the microstructures and morphologies of PPy and PPy–Ag/Cu NPs on the surface of the GCE. As can be seen in Figure 1a, the PPy presents a cauliflower-like form, with each cluster about 2 μm in diameter. With this morphology, the PPy has a large specific surface area and can provide more active sites. After the electrodeposition of Ag and Cu, it can be seen that Ag and Cu are attached to PPy in the form of nanoparticles (Figure 1b). Figure 1c is the image characterized by transmission electron microscopy of PPy–Ag/Cu film stripped from the GCE. It can also be observed that there are many particles on the surface of the film. According to its HRTEM image (Figure 1d), it proves that there is an obvious crystal structure for the particles. By comparing the standard PDF card [24,32], the lattice fringes with a lattice spacing of 0.25 nm and 0.21 nm can be attributed to the (1,1,1) crystal face of Ag and the (1,1,1) crystal face of Cu, respectively. This illustrates that there are Ag and Cu particles in PPy film.

The elements and their distribution of PPy–Ag/Cu are further analyzed by energy dispersive spectrometry (EDS). As shown in Figure 2, after doping Ag and Cu metals, these metal particles are deposited on the surface of the PPy film and evenly distributed along the PPy shape. The SEM image and corresponding EDS images also illustrate the uniformity of the PPy–Ag/Cu film, and the Ag and Cu elements are evenly distributed in the film (Appendix A). Table 1 presents the approximate element contents in the PPy–Ag/Cu film. It can be found that the content of N in PPy–Ag/Cu film is 3.07%, and the contents of Ag and Cu are 2.04% and 20.09%, respectively. It can be speculated that such a structure can effectively overcome the shortcomings of PPy itself, which is not strong in conductivity and poor selectivity.

### 3.2. Optimization of Experimental Parameters

In order to obtain better sensor performance, the experimental parameters were systematically optimized. Firstly, the PPy electropolymerization voltage is optimized. The lower limit of the polymerization voltage was fixed at −1.0 V, and the upper limit of the voltage was set at 0.8 V, 0.9 V, 1.0 V, 1.1 V, and 1.2 V, respectively. The response curve of PPy obtained with different polymerization voltage ranges for 1 mM H_2_O_2_ is shown in Figure 3a. The ΔCurrent can be calculated using the following:ΔCurrent = *C*_2_ − *C*_1_(1)
where *C*_1_ is the average value of the equilibrium current before adding H_2_O_2_, and *C*_2_ is the average value of the equilibrium current after adding 1 mM H_2_O_2_.

Based on Figure 3a, it can be seen that the PPy obtained at a voltage range of −1.0–1.0 V has a maximum response to H_2_O_2_, demonstrating that the PPy has the highest sensitivity to H_2_O_2_ response at this time. This is because the polymerization reaction of polypyridine cannot take place effectively under low voltage (its upper limit is <1.0 V), but PPy obtained from the polymerization reaction of polypyridine will deactivate as its upper limit is over 1.0 V [21]. Meanwhile, the number of cycles in the polymerization process was also optimized. The number of polymerization cycles was set to 7, 8, 9, 10, and 11, respectively, and the response curve of the resulting PPy to 1 mM H_2_O_2_ is presented in Figure 3b. When the number of cycles is 8, the best response performance is obtained. By further increasing the number of cycles to 9 or greater, the response performance is significantly reduced. It seems that when the number of polymerization circles is too large, the PPy layer will become thicker, which will affect the sensitivity performance of the material. So, the polymerization voltage range and the number of cycles of polypyridine are −1.0–1.0 V and eight times, respectively.

A mixture solution of AgNO_3_ and Cu (NO_3_)_2_ is the electrodeposition solution. The metals are loaded onto the surface of PPy to obtain the PPy–Ag/Cu electrode by electrodeposition. Figure 4a,b, respectively, shows the influence of the voltage upper limit and cycle number on the sensing performance in the electrodeposition process. It can be found that the PPy–Ag/Cu obtained at a voltage range of −1.0–1.0 V and ten cycles has the best response to H_2_O_2_. Figure 4c shows the effect of the molar ratio of Cu to Ag in the electrodeposition mixed solution on the sensing performance. It displays that the optimal molar ratio of Ag to Cu is 2. Figure 4d shows the influence of the concentration of the AgNO_3_ and Cu (NO_3_)_2_ mixture on the sensing performance of the modified electrode. The horizontal coordinate is based on the concentration of Ag^+^. The concentration of Cu^2+^ is twice that of Ag^+^. Figure 4d shows that the best concentrations of Ag^+^ and Cu^2+^are 2.5 and 5.0 mM, respectively. Figure 4e presents the response current of different PPy-based electrodes for 1 mM H_2_O_2_. From Figure 4e, it can be seen that the response of the PPy–Ag/Cu electrode is larger than that of PPy, PPy–Ag, and PPy–Cu, which proves that the synergistic effect between Ag and Cu enhances the response of the electrode to H_2_O_2_.

The pH value of the environment has a great influence on the response. The effect of pH on the measurement of the PPy–Ag/Cu electrode was carried out. Figure 4f shows the response curve in the pH range of 5.5–8. It indicates that the response intensity is enhanced by increasing the pH value from 5.5 to 6.5, and the response intensity reaches its highest when the pH value is 6.5; further raising the pH value over 6.5, the response intensity decreases. Therefore, the pH value is set to 6.5 (all the raw data in Figure 4 can be seen in Appendix A and Appendix A in Appendix A).

Then, a series of sensing properties of the PPy–Ag/Cu electrode were characterized. The CV curves are obtained on the same electrode under different concentrations of H_2_O_2,_ as shown in Figure 5a. It can be observed that the response current increases with an increase in H_2_O_2_ concentration. Comparing the CV data with different concentration and time measurements (Appendix A), it can be found that the maximum relative standard deviation of the reduction peak current measurements is below 0.24%. At the same time, the relative standard deviation of reduction peak current for 50 cycles in PBS buffer containing 0.5 mM H_2_O_2_ at a sweep speed of 25 mV/s is 0.19%, and the maximum transfer of reduction peak potential is 0.020 V with a relative standard deviation of 0.6% (Figure 5b), which proves that the PPy–Ag/Cu electrode has good cycle stability. Figure 5c displays these CV curves of the PPy–Ag/Cu electrode in a solution containing 1 mM H_2_O_2_ at different scanning rates (10, 20, 30, 40, 50, 60, 70, 80, 90, and 100 mV/s), showing that the redox peak current is enhanced with increasing scanning rate. Figure 5d shows the relationship between the peak current difference and the square root of the scan rate, illustrating that there is a linear relationship between them. It indicates that the reduction in H_2_O_2_ is a typical diffusion control process.

The linear range of the PPy–Ag/Cu electrode was tested. Figure 6a shows the current response to 1 mM H_2_O_2_ under different voltages. It can be found that when the voltage is set at −0.1 V or −0.3 V, the response is small, which is not conducive to the observation of dataset changes. When the voltage is −0.7 V, excessive noise will cause great fluctuations in the image, which will also affect the measurement of data. Hence, −0.5 V is chosen as the experimental voltage. The *i–t* curve is plotted for the PPy–Ag/Cu electrode at a constant voltage of −0.5 V, and the interval for adding H_2_O_2_ is 60 s (Figure 6b). It is observed that the PPy–Ag/Cu electrode has two linear ranges. The linear range of the first section is 0.1–1.0 mM, corresponding to 0.9978 of the correlation coefficient (Figure 6c). The linear range of the second section is between 1.0 mM and 37.0 mM, corresponding to 0.9969 of the correlation coefficients (Figure 6d). The sensitivity of the first and second linear ranges is 265.06 and 445.02 μA/ (mM × cm^2^), corresponding to the limits of detection (LOD) of 0.027 μM and 0.063 μM, respectively (the signal-to-noise ratio is 3). In comparison with the linear range of the PPy–Ag electrode and the PPy–Cu electrode under the same conditions (Figure 6e,f), it shows that the linear range and sensitivity of the PPy–Ag/Cu electrode are all better than those of the PPy–Ag electrode and the PPy–Cu electrode. It also demonstrates that the co-modification of Ag and Cu can significantly enhance the sensor performance of the electrode. The result of repeating five times dropping 1 mM H_2_O_2_ at the same PPy–Ag/Cu electrode presents a standard deviation of its response current of 0.212, indicating that the PPy–Ag/Cu electrode has good reproducibility.

Table 2 is a comparison of the sensing performances of H_2_O_2_ electrochemical sensors reported in the literature. It can be seen from the comparison that the linear range width, detection limit, and other performance parameters of this work have certain advantages. At the same time, due to the use of metals such as Ag and Cu, which are cheaper than Au, Pt, and Pb, the cost of fabrication of the PPy–Ag/Cu electrode is also largely reduced.

The anti-interference performance and stability of the PPy–Ag/Cu electrode are tested, respectively, as shown in Figure 7. In the anti-interference test, the interfering substances used are citric acid, glucose, and dopamine, respectively. From Figure 7a, it can be seen that the PPy–Ag/Cu electrode has good anti-interference. In addition, the same PPy–Ag/Cu electrode is repeatedly measured for 60 days, and the result displays that the response to 1 mM H_2_O_2_ still maintains the initial value of 89% after 60 days (Figure 7b). Based on the SEM images before and after the experiment (Appendix A), it can be seen that the morphology of the Ag/Cu particles on the PPy film remains stable at the end of the stability test, which ensures that the electrode has good stability. Repeat experiments five times with the same PPy–Ag/Cu electrodes at 1 mM H_2_O_2_ were measured under the same conditions. The standard deviation of its response is 1.126, indicating that the electrode has good repeatability.

Finally, the recovery rates of the PPy–Ag/Cu electrode are shown in Table 3. Three concentrations are selected in each of the two linear ranges, and their recovery rates are calculated. It can be seen that in the first linear range, the recovery rate is 99.84–103.00%. In the second linear range, the recovery is 98.65–104.80%, implying that the PPy–Ag/Cu electrode prepared in this work has a good recovery and a certain practicability.

## 4. Conclusions

In our paper, a kind of H_2_O_2_ electrochemical sensor modified by PPy–Ag/Cu is designed and prepared by electropolymerization and electrodeposition. The results of SEM and TEM analysis demonstrate that Ag and Cu exist in the form of nanoparticles and are uniformly distributed on the PPy film. For detecting H_2_O_2_ in solution, the electrochemical analysis illustrates that the PPy–Ag/Cu sensor has excellent anti-interference, reproducibility, and stability. Its linear range spans from 0.1 mM to 1.0 mM and from 1.0 mM to 35.0 mM, corresponding to detection limits of 0.027 μM and 0.063 μM, respectively. This study provides a simple and easy method for the preparation of nanocomposite electrochemical sensors. At the same time, by reducing the cost, it has a lot of help in promoting its practical application. In the future, the PPy–Ag/Cu composite can also be easily integrated into microelectrodes and implantable and wearable biosensors to detect H_2_O_2_.

## Figures and Tables

**Figure 1 sensors-23-08536-f001:**
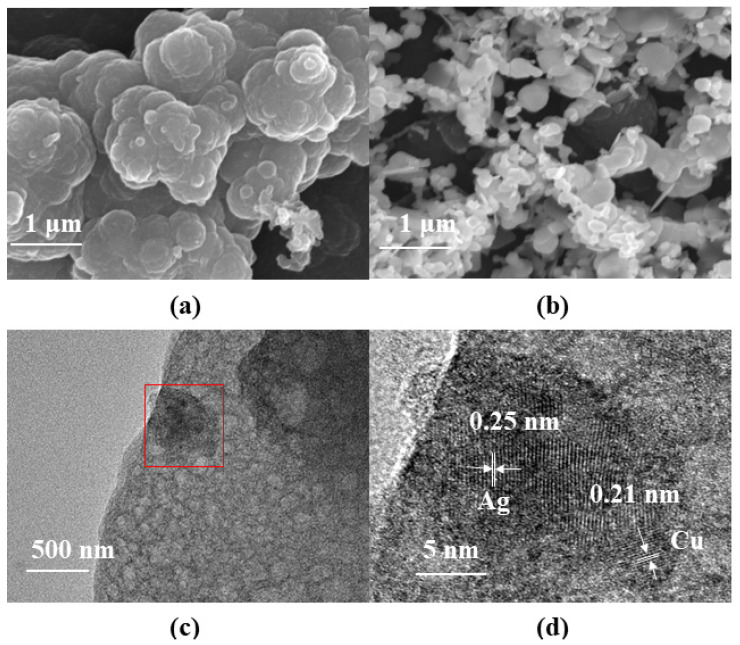
Surface characterization of the modified electrode: (**a**) SEM image of PPy; (**b**) SEM image of PPy–Ag/Cu film; (**c**) TEM image of PPy–Ag/Cu film; (**d**) HRTEM image of PPy–Ag/Cu film (red box part of Figure 1c).

**Figure 2 sensors-23-08536-f002:**
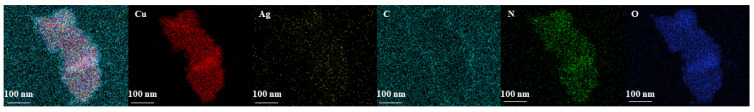
EDS image of PPy–Ag/Cu.

**Figure 3 sensors-23-08536-f003:**
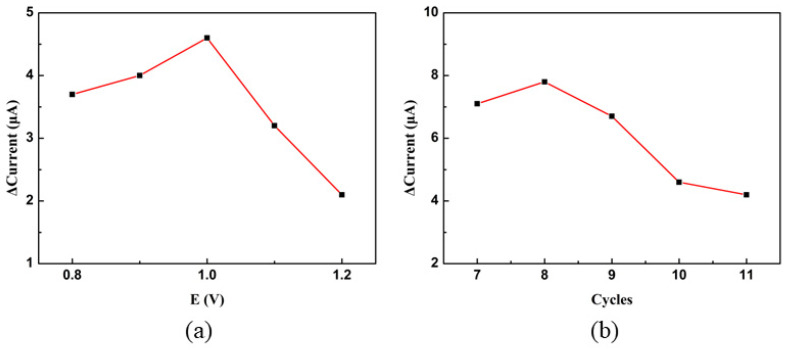
(**a**) The effect of the upper voltage limit of the polypyridine polymerization process on the sensing performance. (**b**) The effect of the number of polymerization cycles on the sensing performance.

**Figure 4 sensors-23-08536-f004:**
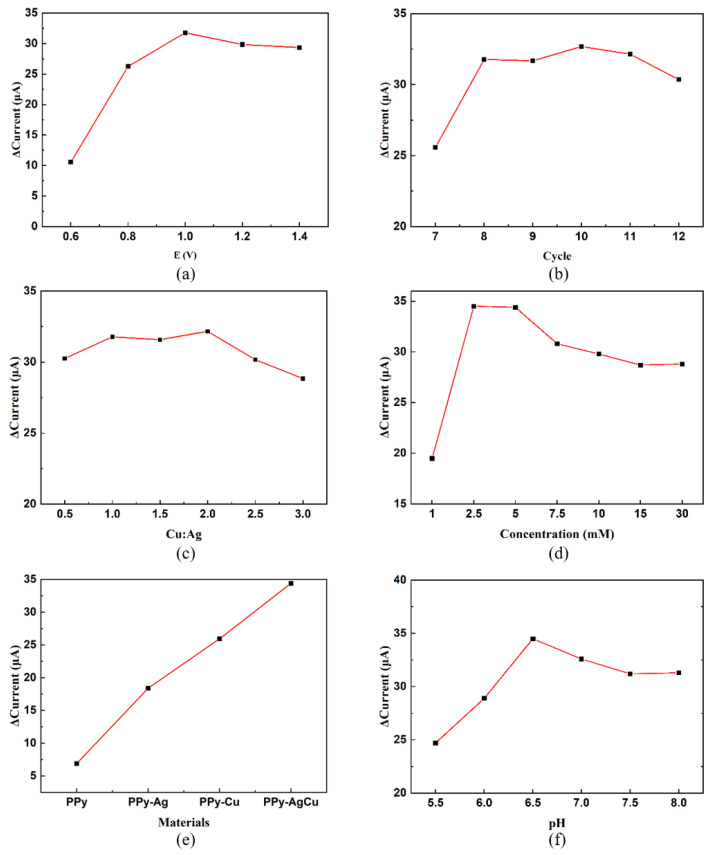
(**a**) Influence of the upper voltage limit during Ag and Cu electrodeposition on sensing performance. (**b**) Influence of electrodeposition cycles on sensing performance. (**c**) The effect of the molar ratio of Cu to Ag on the sensing performance (Ag concentration was fixed at 5 mM). (**d**) Response of PPy electrode obtained from electrodeposited in solution with different Ag concentrations (mM, Cu:Ag as 2:1). (**e**) Response of electrodes modified with different materials (all materials are at optimal concentrations for performance). (**f**) Effect of different pH values on the response.

**Figure 5 sensors-23-08536-f005:**
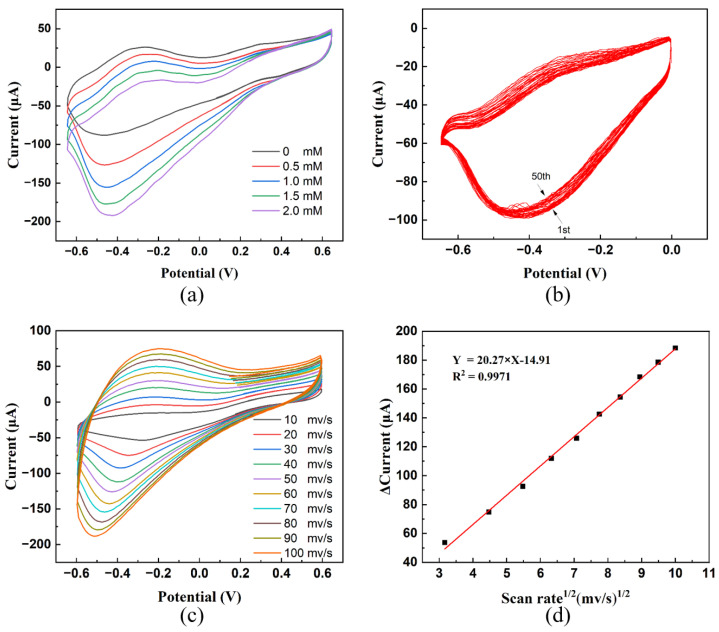
(**a**) CV curves of the PPy–Ag/Cu electrode under different concentrations of H_2_O_2_. (**b**) Cyclic stability curve of the PPy–Ag/Cu electrode at 1 mM H_2_O_2_. (**c**) CV curves of the PPy–Ag/Cu electrode in PBS solution at different scanning rates. (**d**) The relationship between the reduction peak current and the square root of the scanning rate.

**Figure 6 sensors-23-08536-f006:**
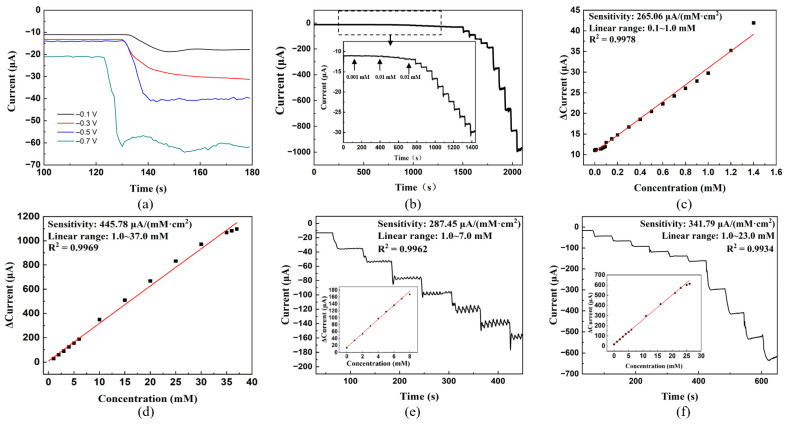
Measurement of sensing linear range of PPy–Ag/Cu, PPy–Ag, PPy–Cu electrodes: (**a**) current response to H_2_O_2_ at different voltages; (**b**) *i–t* curve of PPy–Ag/Cu electrode (−0.5 V, 0.2 M PBS solution with pH = 6.5); (**c**) *i–t* fitting curve for a concentration range of 0.05–1.0 mM (PPy–Ag/Cu electrode). (**d**) *i–t* fitting curve for a concentration range of 1.0–35.0 mM (PPy–Ag/Cu electrode). (**e**) *i–t* curve of the PPy–Ag electrode. (**f**) *i–t* curve of the PPy–Cu electrode.

**Figure 7 sensors-23-08536-f007:**
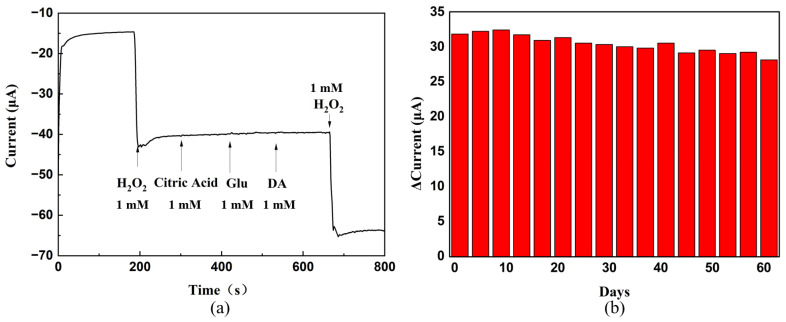
(**a**) Anti–jamming test of the PPy–Ag/Cu electrode. (**b**) Stability of the PPy–Ag/Cu electrode.

**Table 1 sensors-23-08536-t001:** Element composition of PPy–Ag/Cu sensitive film.

C	N	O	Cu	Ag
40.76%	3.07%	31.70%	20.09%	2.04%

**Table 2 sensors-23-08536-t002:** Performance comparison with different electrochemical H_2_O_2_ sensors.

Materials	Linear Range	Sensitivity	LOD *	Ref.
PPy–GO–AuNPs	2.5–25 mM	41.35 μA/ (mM × cm^2^)	5 μM	[20]
PPy–rGO–Au	0.032–2 mM	317 μA/ (mM × cm^2^)	2.7 μM	[21]
PPy–PtPd NP	2.5–400 μM	1360.83 μA/ (mM × cm^2^)	2.5 μM	[18]
PPy–Cu	0.4–1 mM	-	0.51 μM	[32]
	1–12 mM	510 μA/ (mM × cm^2^)	4.39 μM	
CMC/PPy/PB	20–1100 μM	456.8 μA/ (mM × cm^2^)	5.23 μM	[39]
AgNPs–TWEEN–GO	0.02–23.1 mM	-	8.7 μM	[40]
CoFe_2_O_4_/CNTs	5–50 μM	-	0.05 μM	[41]
AgNSs	5–6000 μM	-	0.17 μM	[42]
PPy–Ag/Cu	0.1–1.0 mM 1.0–37.0 mM	265.06 μA/ (mM × cm^2^) 445.78 μA/ (mM × cm^2^)	0.027 μM 0.063 μM	This work

* LOD = 3*σ*/S, where S is the slope of the calibration curve, and σ is the standard deviation of the blank solution.

**Table 3 sensors-23-08536-t003:** The recovery rate of the PPy–Ag/Cu electrode to H_2_O_2_.

Linear Range	Added H_2_O_2_ Concentration (mM)	Measured H_2_O_2_ Concentration (mM)	Measured H_2_O_2_ Mean Concentration (mM)	Recovery (%)
0.1~1	0.2000	0.2035	0.2048	102.40
		0.1983		
		0.2127		
	0.5000	0.5311	0.5150	103.00
		0.5043		
		0.5097		
	0.8000	0.8003	0.7987	99.84
		0.7914		
		0.8044		
1~35	5.000	4.9724	4.9323	98.65
		4.8834		
		4.9412		
	15.00	15.5333	15.7207	104.80
		15.4646		
		16.1641		
	25.00	26.3112	26.1462	104.58
		26.0541		
		26.0733		

## Data Availability

Not applicable.

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
