# Peer review of "Hydrogen Peroxide Electrochemical Sensor Based on Ag/Cu Bimetallic Nanoparticles Modified on Polypyrrole"

_sensors, 2023, doi:10.3390/s23208536_

Round 1
Reviewer 1 Report
Please find the Word document for the comments.

The manuscript is well-written, and I do not have any difficulty understanding this manuscript.
Reviewer 2 Report
The manuscript “Hydrogen Peroxide Electrochemical Sensor Based on Ag/Cu Bimetallic Nanoparticles Modified on Polypyrrole” reports the preparation of PPy-Ag/Cu composite material by electropolymerization and electrodeposition for the detection of H2O2. The results demonstrate that the PPy-Ag/Cu disclosed good sensing performance and anti-interference ability. In addition, the preparation cost can be reduced to a certain extent. Based on the novelty and achieved sensing performances, I recommend publication of this manuscript in Sensors after minor revisions according the following comments.
Comments:
1. Please unify the sizes of Figure 5 and Figure 6.
2. How about the relative sensing performances of Ag, Cu or Ag/Cu without PPy?
3. The format of the references, like the reference [28], is not identical with others.
4. Some spelling errors should be corrected. Please carefully check the spelling, grammar and so on in this paper.
A minior language polishing is suggested.
Reviewer 3 Report
The authors report the preparation of an electrochemical sensor for the detection of hydrogen peroxide based on electropolymerized polypyrrole with Ag/Cu nanoparticles (PPy-Ag/Cu electrode). They have characterized the NP deposited on the polymer by SEM and HRTEM. They have optimized the conditions for the preparation of the electrodeposited polymer for the detection of hydrogen peroxide. Although modified-electrodes based on electropolymerized polypirrole have been widely investigated for the fabrication of electrochemical sensors, the reported polypyrrole-modified electrode is relevant for the development of H2O2 sensors with the future prospective to integrate this system into microelectrodes for wearable sensors. For the most part, the prepared materials and the developed sensor have been well characterized. Nonetheless, the manuscript present some issues that need to be addressed.
1) The rational for the implementation of Ag/Cu nanoparticles for the detection of hydrogel peroxides should be discussed in the Introduction. In addition, the methods for the Ag/Cu preparation, by using electrodeposition, should be briefly described and discussed in the Introduction and references should be cited, for example:
- Nanoporous Copper–Silver Alloys by Additive-Controlled Electrodeposition for the Selective Electroreduction of CO2 to Ethylene and Ethanol. doi.org/10.1021/jacs.8b01868
- Structural properties of electrodeposited Cu-Ag alloys. doi.org/10.1016/j.electacta.2017.08.097.
2) The authors should discuss in the Results and Discussion section why the sensing performance of the modified electrode decreased after 8 cycles of electropolymerization (Figure 3b).
3) The stability of the PPy-Ag/Cu electrode in detecting H2O2 is one of the advantage of this sensor, as showed in Figure 7B. The authors should demonstrate that the morphology of the Ag/Cu particles on the polypyrrole should remain stable at the end of the stability test. On this regard, they should provide the characterization of the surface properties of the modified electrode after 60 days by SEM.
4) The approach used for the determination of the LOD should be reported in the manuscript.
5) Since the authors are presenting their work in the context of generating catalytic materials integrated with polymers for the detection of H2O2, nanozymes integrated with polymers for the optical or electrochemical detection of H2O2 have been prepared by using mild conditions. A brief comment about these approaches should be included in the introduction, and the following references should be cited:
- GOx@ZIF-8(NiPd) nanoflower: an artificial enzyme system for tandem catalysis. doi.org/10.1002/anie.201710418.
- Nanozyme–cellulose hydrogel composites enabling cascade catalysis for the colorimetric detection of glucose. doi.org/10.1021/acsanm.2c01609
- Platinum nanozyme-hydrogel composite (PtNZHG)-impregnated cascade sensing system for one-step glucose detection in serum, urine, and saliva. doi.org/10.1016/j.snb.2022.131585.
The manuscript required only moderate edition of the language.
